# An Atlas of the Quantitative Protein Expression of Anti-Epileptic-Drug Transporters, Metabolizing Enzymes and Tight Junctions at the Blood–Brain Barrier in Epileptic Patients

**DOI:** 10.3390/pharmaceutics13122122

**Published:** 2021-12-09

**Authors:** Risa Sato, Kotaro Ohmori, Mina Umetsu, Masaki Takao, Mitsutoshi Tano, Gerald Grant, Brenda Porter, Anthony Bet, Tetsuya Terasaki, Yasuo Uchida

**Affiliations:** 1Graduate School of Pharmaceutical Sciences, Tohoku University, Sendai 980-8578, Japan; risa.sato.t8@dc.tohoku.ac.jp (R.S.); omori.dds.2018@gmail.com (K.O.); ume081812@gmail.com (M.U.); tetsuya.terasaki.d5@tohoku.ac.jp (T.T.); 2Department of Neurology and Brain Bank, Mihara Memorial Hospital, Isesaki 372-0006, Japan; msktakaobrb@gmail.com (M.T.); mmh-byori@mihara-ibbv.jp (M.T.); 3Department of Clinical Laboratory, National Center of Neurology and Psychiatry, National Center Hospital, Kodaira 187-8551, Japan; 4Department of Neurosurgery, Stanford University, Stanford, CA 94305, USA; ggrant2@stanford.edu (G.G.); abet0915@stanford.edu (A.B.); 5Department of Neurology, Stanford University, Stanford, CA 94305, USA; brenda2@stanford.edu

**Keywords:** blood–brain barrier, epilepsy, anti-epileptic drug, transporter, enzyme, tight junction, SWATH

## Abstract

The purpose of the present study was to quantitatively elucidate the levels of protein expression of anti-epileptic-drug (AED) transporters, metabolizing enzymes and tight junction molecules at the blood–brain barrier (BBB) in the focal site of epilepsy patients using accurate SWATH (sequential window acquisition of all theoretical fragment ion spectra) proteomics. Brain capillaries were isolated from focal sites in six epilepsy patients and five normal brains; tryptic digests were produced and subjected to SWATH analysis. MDR1 and BCRP were significantly downregulated in the epilepsy group compared to the normal group. Out of 16 AED-metabolizing enzymes detected, the protein expression levels of GSTP1, GSTO1, CYP2E1, ALDH1A1, ALDH6A1, ALDH7A1, ALDH9A1 and ADH5 were significantly 2.13-, 6.23-, 2.16-, 2.80-, 1.73-, 1.67-, 2.47- and 2.23-fold greater in the brain capillaries of epileptic patients than those of normal brains, respectively. The protein expression levels of Claudin-5, ZO-1, Catenin alpha-1, beta-1 and delta-1 were significantly lower, 1.97-, 2.51-, 2.44-, 1.90- and 1.63-fold, in the brain capillaries of epileptic patients compared to those of normal brains, respectively. Consistent with these observations, leakage of blood proteins was also observed. These results provide for a better understanding of the therapeutic effect of AEDs and molecular mechanisms of AED resistance in epileptic patients.

## 1. Introduction

Epilepsy is one of the most common central nervous system (CNS) disorders, with an incidence of approximately 1% of the population. Post-traumatic epilepsy as a comorbidity of traumatic brain injuries is also becoming a universal challenge for brain health due to the increasing incidence of brain trauma [1]. About one third of patients eventually develop drug resistance to some anti-epileptic drugs (AEDs) and fail to respond to treatment with medication [2], although a variety of combination pharmacotherapy of multiple drugs was investigated [3]. The blood–brain barrier (BBB) regulates the penetration of AEDs into the CNS by forming tight junctions and by expressing drug transporters and drug-metabolizing enzymes. Quantitative analysis of the expression and function of transporters, enzymes and tight junction proteins at the BBB in epilepsy patients is important in terms of understanding the therapeutic effects of AEDs, since their penetration is an important factor affecting the therapeutic efficacy of such drugs.

It has been reported that the P-glycoprotein (P-gp/MDR1), which is responsible for the efflux of AEDs, is overexpressed in 11 out of 19 epilepsy patients with mRNA expression [4]. However, the MDR1 inhibitor verapamil, when administered in combination with AEDs, is not effective in improving the frequency of epileptic seizures [5], and therefore the question remains as to whether MDR1 is involved in AED resistance. At the protein level, immunostaining has shown that MDR1 expression is upregulated in cerebral blood vessels [6], but antibody-based analysis is not sufficiently quantitative to confirm this with certainty. Similar to MDR1, the breast cancer resistance protein (BCRP/ABCG2) is an efflux transporter of a variety of AEDs [7]. While not widely analyzed for BCRP unlike MDR1, antibody-based analyses suggest that BCRP expression at the BBB is not altered in epilepsy patients [6].

The BBB expresses a variety of drug-metabolizing enzymes, including cytochrome 2E1 (CYP2E1) and glutathione S-transferase (GST) [8,9]; e.g., CYP2E1 metabolizes ethosuximide, felbamate, phenobarbital and carbamazepine [10]. GST metabolizes phenytoin, valproate and carbamazepine [10,11,12]. It is noteworthy that the absolute protein abundance of GSTP1 in the human BBB is about six times higher than that of MDR1 and about four times higher than that of BCRP [9]. It is therefore possible that it might contribute significantly to the deactivation of AEDs. However, for metabolic enzymes, variations in the extent of expression at the protein level in the BBB of epilepsy patients have not been elucidated yet.

The integrity of the tight junctions at the BBB is also a factor that influences the brain permeability of AEDs. It has been reported that the systemic administration of pentylenetetrazol or kainic acid increases the infiltration of albumin into the brain in rabbit and rat models of epilepsy [13,14]. Immunohistochemical staining has also shown that albumin infiltrates into the brain in focal areas of epilepsy patients [13]. As a mechanism for BBB disruption, it has been reported that the expression of tight junction proteins such as ZO-1, Occludin, Claudin-1 and Claudin-5 is decreased at the BBB in epileptic rat models [15]. However, it is not known to what extent the expression of these tight junction molecules is reduced at the BBB of epilepsy patients, or whether the tight junction is disrupted to the same extent as at the BBB of epileptic rat models.

The SWATH (sequential window acquisition of all theoretical fragment ion spectra) method is one of the more recent comprehensive quantitative proteomics methods that have been established, and its quantitative accuracy is excellent compared with previous comprehensive proteomics, which give it a significant advantage [16]. Multiple specific peptides derived from each protein can be quantified, and change in the level of protein expression of the target protein can be accurately quantified based on the average of these peptides. Membrane proteins contain hydrophobic regions, such as transmembrane sites, which result in their incomplete solubilization and resistance to trypsin digestion. However, we were able to improve the accuracy of the SWATH method by completely solubilizing such proteins with guanidine hydrochloride, thus improving the efficiency of the tryptic digestion of membrane proteins [17], and by applying in silico peptide selection criteria [18], such as excluding transmembrane sites and sequences that are more resistant to tryptic digestion from the numerous peptides that are measured [19,20]. Therefore, the low quantitative accuracy of conventional antibody-based quantitative analysis in epilepsy BBB studies would be predicted to be significantly improved.

The purpose of the present study was to quantitatively elucidate the levels of protein expression of AED transporters, metabolizing enzymes and tight junction molecules at the BBB in epileptic patients using this accurate SWATH method.

## 2. Materials and Methods

### 2.1. Human Brain Tissues

Age, gender, AED treatment, brain region and pathological diagnosis of the subjects are summarized in Table 1. The focal sites in the brains of the epileptic patients were provided by the department of Neurosurgery, Stanford University. Normal brain tissues were provided from the department of Neurology and Brain Bank, Mihara Memorial Hospital or purchased from Analytical Biological Services (Wilmington, DE, USA). The protocol in the present study was approved by the ethical committees in all institutes. The details can be found in the “Institutional Review Board Statement” section below.

### 2.2. Isolation of Brain Capillaries from Human Brains

The brain capillaries were isolated as previously described [21,22]. The representative microscopic images of the isolated brain capillaries were shown in Appendix A, indicating that the purity of the isolated brain capillaries was not significantly different among epileptic focal site, control cortex and control white matter groups.

### 2.3. Sample Preparation for SWATH-Based Quantitative Proteomics

Protein digestion was performed as described previously [23]. The tryptic digests were cleaned up with a self-packed SDB-XD 200 µL tip (3M, Maplewood, MN, USA) as previously described [24].

### 2.4. LC-MS/MS Measurement for SWATH-Based Quantitative Proteomics

The cleaned peptide samples were injected into an NanoLC 425 system (Eksigent Technologies, Dublin, CA, USA) coupled with an electrospray-ionization Triple TOF 5600 mass spectrometer (SCIEX, Framingham, MA, USA), which was set up for a single direct injection and analyzed by SWATH-MS acquisition, as previously described [20].

### 2.5. Data Analysis for SWATH-Based Quantitative Proteomics

Spectral alignment and data extraction from the SWATH chromatogram were performed with the SWATH Processing Micro App (SCIEX, Toronto, Canada) in Peakview (SCIEX) using in-house spectral libraries as previously described [20]. The parameters for peak data extraction by Peakview were as follows: Number of Peptides per Protein, 999; Number of transitions per Peptide, 6; Peptide Confidence Threshold, 99%; False Discovery Rate (FDR) Threshold, 1.0%; XIC Extraction Window, ±4.0 min; XIC width (ppm), 50. According to the procedure previously described [19], unreliable peaks and peptides were removed based on the criteria of data selection and amino acid sequence-based peptide selection, and the peak areas at the peptide level were calculated as an average of those in the transition level after normalizing the differences in signal intensity between the different transitions. The details have been reported in our previous study [19]. The peak areas of individual proteins were calculated as an average of those at the peptide level, and were compared between epileptic patients and normal groups.

### 2.6. Statistical Analysis

All statistical analyses were performed under the null hypothesis, assuming that the means for the compared groups were equal. Comparison between two groups was performed by an unpaired two-tailed Student’s *t*-test (equal variance) or Welch’s test (unequal variance) according to the result of the F test, and followed by a Benjamini–Hochberg (BH) correction. If the *p*-value was less than 0.05, the difference was considered to be statistically significant and the null hypothesis was rejected. No formal power calculation was performed to estimate the required sample size. No randomization or blinding was performed in this study.

## 3. Results

### 3.1. Protein Expression Levels of AED Transporters in Brain Capillaries

Three transporters (MDR1, BCRP and MCT8) and LAT1 are involved in the efflux and influx of AEDs at the BBB, respectively (Table 2) [7,25,26]. The brain capillaries were isolated from the focal sites of epileptic brains or normal brains, and subjected to the SWATH analysis. A total of 2964 proteins were quantified and compared between the epilepsy and normal brain groups (Appendix A). The protein expression levels of MDR1 and BCRP were significantly smaller in the epilepsy group than in the normal group. MCT8 and LAT1 were not significantly different between the two groups (Figure 1).

### 3.2. Protein Expression Levels of AED-Metabolizing Enzymes in Brain Capillaries

The relationship between AEDs and metabolizing enzymes are listed in Table 2. 16 AED-metabolizing enzymes (GSTP1, GSTO1, GSTM3, GSTK1, MGST2, CYP2E1, ALDH1A1, ALDH2, ALDH3A2, ALDH6A1, ALDH7A1, ALDH9A1, ADH5, EPHX1, CBR1 and CBR3) were detected in the brain capillaries (Figure 2). Of these enzymes, the protein expression levels of GSTP1, GSTO1, CYP2E1, ALDH1A1, ALDH6A1, ALDH7A1, ALDH9A1 and ADH5 were significantly greater, 2.13-, 6.23-, 2.16-, 2.80-, 1.73-, 1.67-, 2.47- and 2.23-fold in the brain capillaries of epileptic patients than those of normal brains, respectively (Figure 2).

### 3.3. Protein Expression Levels of Tight Junction Molecules in Brain Capillaries

Decreased expression levels of tight junction molecules weaken the integrity of tight junction at the BBB, which would be predicted to increase the permeability of AEDs into brain. As shown in Figure 3, the protein expression levels of Claudin-5, ZO-1, Catenin alpha-1, beta-1 and delta-1 were significantly 1.97-, 2.51-, 2.44-, 1.90- and 1.63-fold smaller in the brain capillaries of epileptic patients than those of normal brains, respectively (Figure 3). This suggests that the protein expression of major tight junction proteins are downregulated at the BBB in the focal sites of epileptic patients.

### 3.4. Infiltration of Blood Proteins

If the tight junction is weakened, blood proteins can infiltrate to the lateral spaces of endothelial cells or the brain-side surface of endothelial cells, resulting in increased amounts of blood proteins being in the isolated brain capillary fraction. As shown in Figure 4, the levels of the hemoglobin subunit delta, alpha, beta, fibrinogen beta chain, gamma chain, serum albumin, fibrinogen alpha chain and complement C3 in the brain capillary fractions were significantly greater (26.1-, 18.7-, 18.3-, 8.96-, 8.70-, 8.48-, 7.02- and 5.34-fold) in epileptic patients compared to those of normal brains, respectively (Figure 4). This suggests that the blood proteins infiltrate into the lateral spaces of endothelial cells or the brain-side surface of endothelial cells in epilepsy patients.

## 4. Discussion

The protein expression of AED-related proteins in cerebral vessels of epilepsy patients has been quantified by immunohistochemical analysis [6,37]. The detection of proteins in human tissue sections such as formalin-fixed paraffin-embedded tissue section by immunohistochemistry is semi-quantitative. The accurate and robust quantification of protein expression is a challenging task [38,39]. While antibody-based analysis with high quantitative accuracy has recently emerged, the quantitative analysis of cerebral blood vessels in epilepsy described above has been performed only by conventional immunohistochemical analysis, and an accurate quantitative analysis of the BBB in epilepsy patients has not yet been reported. In contrast, it was reported that disease-associated changes and differences in protein expression levels, as quantified by quantitative proteomics, accurately reflects differences in the in vivo transport activity of the transporter [40,41,42]. Therefore, in the present study, we analyzed AED transporters, enzymes and tight junction molecules at the BBB using the SWATH method, which has excellent quantitative accuracy, and then quantitatively clarified changes in their expression at the BBB in epilepsy patients for the first time. Unlike antibody-based analyses, the level of expression of the MDR1 protein was found to be decreased in epilepsy subjects (Figure 1). BCRP levels were also decreased. Other AED transporters (MCT8 and LAT1) were unchanged (Figure 1). The expression of various AED metabolizing enzymes in the human BBB is reported for the first time, and the protein expression levels of GSTP1, GSTO1, CYP2E1, ALDH1A1, ALDH6A1, ALDH7A1, ALDH9A1 and ADH5 were found to be significantly increased at the BBB in the epileptic focus (Figure 2). These metabolic enzymes may be involved in the tolerance of AEDs. Regarding tight junction molecules, the protein expression levels were decreased at the BBB of epileptic patients (Figure 3), as has also been reported in animal models of epilepsy [13,14]. In line with this, the leakage of blood proteins was also shown (Figure 4).

MDR1 at the BBB has been reported to be involved in the resistance to AEDs [4]. In contrast, the MDR1 inhibitor verapamil, when administered in combination with AEDs, failed to improve the frequency of epileptic seizures [5]. In the present study, the level of expression of MDR1 at the BBB was lower in epileptic patients compared to the BBB of normal subjects (Figure 1), suggesting that MDR1 has little influence on tolerance to AEDs. Furthermore, because of weakened tight junctions (Figure 3 and Figure 4), the MDR1-mediated resistance to AEDs may be less pronounced. MDR1 at the BBB has been thought to be involved in drug resistance, as previous studies in rodents have shown that MDR1 expression limits the brain transfer of antiepileptic drugs. However, in humans, there is little data to suggest that MDR1 is involved in antiepileptic drug resistance. Immunostaining experiments have shown that the signal intensity of MDR1 is increased, but the specificity and quantitative performance of the antibody is limited. In this study, we applied the SWATH method, which has excellent specificity and quantitative accuracy, and for the first time we were able to accurately quantify the expression levels of MDR1 and BCRP. Therefore, unlike rodents, MDR1 and BCRP do not seem to have a significant effect on antiepileptic drug resistance in humans. Tight junctions were weakened in the focal area of human epilepsy, in agreement with reports in rodents [13,14]. Weakened tight junctions allow blood components to infiltrate the brain, and these have been reported to increase the frequency of epileptic seizures [13]. Thus, the brain transfer of antiepileptic drugs may be also increased, but the degree to which they induce convulsive seizures could be also increased, which may apparently indicate a weaker effect of antiepileptic drugs.

Angiogenesis has been reported to occur in the epileptic brain [43]. No significant increase in the levels of expression of the MDR1 protein was detected in the present SWATH analysis (Figure 1), which indicates that the amount (moles) of MDR1 protein expressed per μg of blood vessel protein was not increased. If angiogenesis increases the amount of blood vessels in the brain per unit volume, then the total levels of expression of efflux transporters (MDR1, BCRP and MCT8) in the unit volume would also increase. It is therefore possible that the distribution of various AEDs (Table 2) in the brain could be limited, which would then lead to the pharmaco-resistance for AEDs. As shown in Figure 2, the expression of the ALDH family members was increased. ALDH has been reported to promote angiogenesis [44]. Therefore, angiogenesis may be promoted by an increased expression of the ALDH family of proteins at the epileptic focus. The number of donors was limited in this study (Table 1), and quantitative analyses using further specimens and investigations of vessel density will be required to understand the overall involvement of efflux transporters in the tolerance to AEDs.

As shown in Table 2, a variety of metabolic enzymes are involved in the metabolism of AEDs. The findings reported in this study show that the expressions of GSTP1, GSTO1, CYP2E1, ALDH1A1, ALDH6A1, ALDH7A1, ALDH9A1 and ADH5 are increased in the BBB of epilepsy patients (Figure 2). Because CYP2E1 and GSTs metabolize a range of AEDs, it is possible that their increased expression could affect AEDs resistance. At the normal BBB, GSTP1 is expressed six-fold more abundantly than MDR1, and GSTO1 is expressed in absolute amounts comparable to MDR1, suggesting that GSTP1 and GSTO1 could sufficiently contribute to the substrate metabolism at the human BBB [9]. At the BBB in the epileptic focus, GSTP1 and GSTO1 were upregulated by 2.13- and 6.23-fold, respectively (Figure 2). GSTs are involved in the metabolism of major AEDs such as phenytoin, valproate and carbamazepine (Table 2). Furosemide, an inhibitor of GSTs, has been reported to potentiate the anticonvulsant action of valproate [45]. These findings suggest that GSTP1 and GSTO1 could be involved in the drug resistance of AEDs. In future studies, in vivo analyses will be needed to demonstrate whether the enzymes that are up-regulated at the epileptic BBB (Figure 2) are actually involved in the drug resistance of AEDs.

A limitation of the present study is the small number of donors. Therefore, it cannot be said that all types of epilepsy patients are covered. In terms of age, patients with epilepsy tend to be younger than controls: Control 1 and patients with epilepsy 3, 4 and 6 overlap in age; for the molecules shown in Figure 1, Figure 2, Figure 3 and Figure 4, there is no significant difference in expression between control 1 and older control donors (donors 2, 3 and 4) (Appendix A). Therefore, the quantitative differences between epilepsy patients and controls shown in this study may be due to the pathogenesis of epilepsy. Only epileptic patient 5 was female, and the expression of GSTP1 was lower than in male epileptic patients (Appendix A). GSTP1 is expressed in the liver at lower levels in females than in males [46]. Therefore, in BBB, the amount of GSTP1 may be lower in females. Further studies are needed to increase the number of samples and to equalize the age range and gender between epilepsy and control groups.

We have previously reported that MDR1 protein expression at the BBB is increased in mouse models of epilepsy [41], but in contrast, it is decreased in patients with epilepsy (Figure 1). Table 1 lists the medications that epilepsy patients have been taking. These have not been reported to reduce MDR1 protein expression. While the different age range of the epilepsy and control groups may have an effect, the expression of MDR1 was lower in the epilepsy group when compared between control 1 and epilepsy patients of the same age (donors 3, 4 and 6). In young adulthood, the expression of MDR1 in the BBB is highly variable in mice, but not in primates [47,48]. While the detailed mechanisms are unknown, it may be likely that differences in the regulation of MDR1 expression between mice and primates result in the contrasting changes in expression levels at the BBB in mice and humans with epilepsy.

To investigate whether there are regional differences in the expression levels of the BBB, we isolated blood vessels from white matter and cortex for control 4 and performed SWATH analysis. The results showed that there was no significant difference in the expression of molecules in Figure 1, Figure 2, Figure 3 and Figure 4 between white matter and cortical BBB (Appendix A).

## 5. Conclusions

SWATH-based quantitative proteomics with excellent quantitative accuracy was applied to the analysis of components of the BBB in the focal site of epilepsy patients, and finally gave a quantitative atlas illustrated in Figure 5. The results show that the protein expression levels of MDR1 and BCRP are downregulated, and those of MCT8 and LAT1 are comparable to a normal BBB. The findings also revealed that the protein expression levels of eight AED-metabolizing enzymes (GSTP1, GSTO1, CYP2E1, ALDH1A1, ALDH6A1, ALDH7A1, ALDH9A1 and ADH5) are significantly upregulated at the BBB in the focal site of epilepsy patients, whereas the protein expression levels of tight junction molecules are significantly downregulated and this is accompanied by an increased infiltration of blood proteins. The present study has only revealed protein expression levels and it is not yet known whether these changes in expression levels affect the brain pharmacokinetics and pharmacological effects of antiepileptic drugs. Further experimental justification is required.

## Figures and Tables

**Figure 1 pharmaceutics-13-02122-f001:**
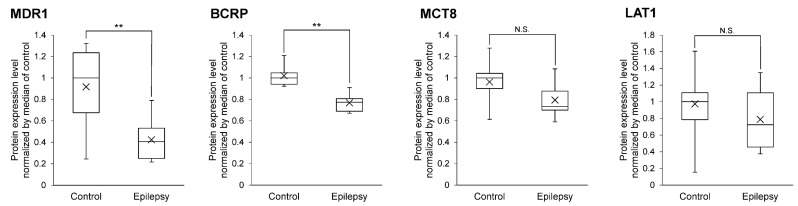
Protein expression level of the AED transporters in epileptic brain capillaries compared to controls. Brain capillaries were isolated from focal sites in six epilepsy patients and five normal brains; tryptic digests were produced and subjected to SWATH analysis. The band inside the box represents the median, and the bottom and top of the box indicate the first and third quartiles, respectively. Whiskers indicate the minimum and maximum values of the protein levels. X plots show the average in each group. The data are normalized by the median for the protein expression level in controls. ** BH-adjusted *p* value < 0.01, significantly different between epilepsy and control groups. N.S., not significantly different. MDR1, Multidrug resistance protein 1; BCRP, Breast cancer resistance protein; LAT1, L-type amino acid transporter 1; MCT8, Monocarboxylate transporter 8.

**Figure 2 pharmaceutics-13-02122-f002:**
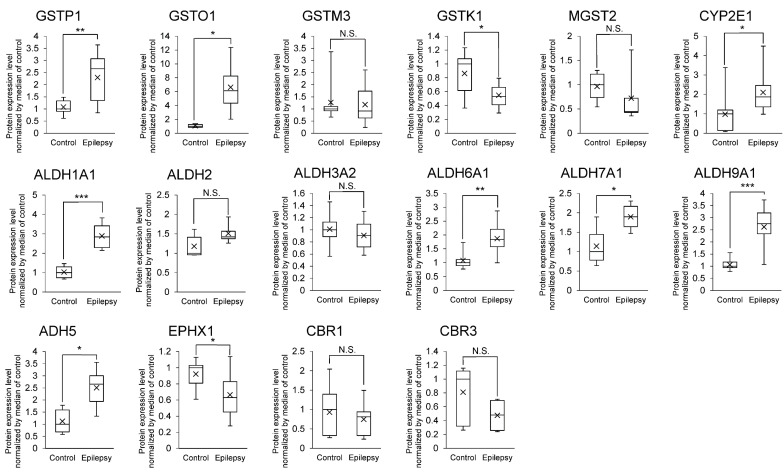
Protein expression level of AED-metabolizing enzymes in epileptic brain capillaries compared to controls. Brain capillaries were isolated from focal sites in six epilepsy patients and five normal brains; tryptic digests were produced and subjected to SWATH analysis. The band inside the box represents the median, and the bottom and top of the box indicate the first and third quartiles, respectively. The whiskers indicate the minimum and maximum values for the protein levels. X plots show the average in each group. The data are normalized by the median of the protein expression level in controls. * BH-adjusted *p* value < 0.05, ** BH-adjusted *p* value < 0.01 and *** BH-adjusted *p* value < 0.001, significantly different between the epilepsy and control groups. N.S., not significantly different. GSTP1, Glutathione S-transferase pi; GSTO1, Glutathione S-transferase omega-1; GSTM3, Glutathione S-transferase Mu 3; GSTK1, Glutathione S-transferase kappa 1; MGST2, Microsomal glutathione S-transferase 2; CYP2E1, Cytochrome P450 2E1; ALDH1A1, Aldehyde dehydrogenase family 1 member A1; ALDH2, Aldehyde dehydrogenase 2; ALDH3A2, Aldehyde dehydrogenase family 3 member A2; ALDH6A1, Aldehyde dehydrogenase family 6 member A1; ALDH7A1, Aldehyde dehydrogenase family 7 member A1; ALDH9A1, Aldehyde dehydrogenase family 9 member A1; ADH5, Alcohol dehydrogenase 5; EPHX1, Epoxide hydrolase 1; CBR1, Carbonyl reductase 1; CBR3, Carbonyl reductase 3.

**Figure 3 pharmaceutics-13-02122-f003:**
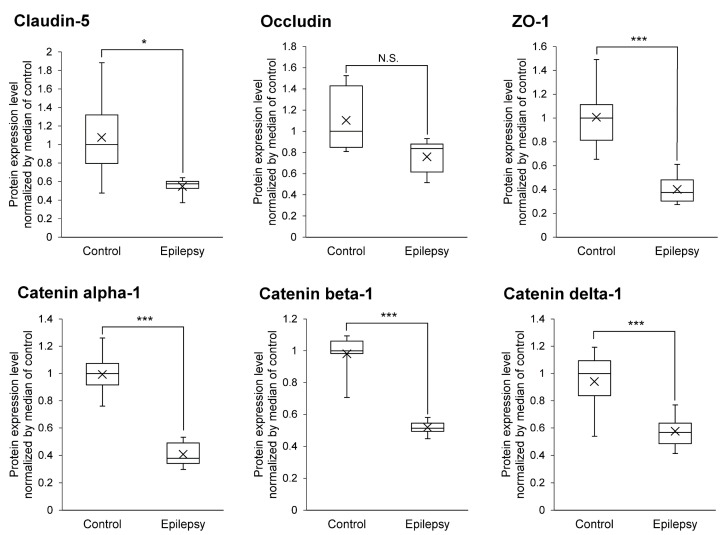
Protein expression level of tight junction and adherence junction proteins in epileptic brain capillaries compared to controls. Brain capillaries were isolated from focal sites in six epilepsy patients and five normal brains; tryptic digests were produced and subjected to SWATH analysis. The band inside the box represents the median, and the box’s bottom and top indicate the first and third quartiles, respectively. The whiskers indicate the minimum and maximum values of the protein levels. X plots show the average for each group. The data are normalized by median of the protein expression level in control. * BH-adjusted *p* value < 0.05 and *** BH-adjusted *p* value < 0.001, significantly different between epilepsy and control groups. N.S., not significantly different.

**Figure 4 pharmaceutics-13-02122-f004:**
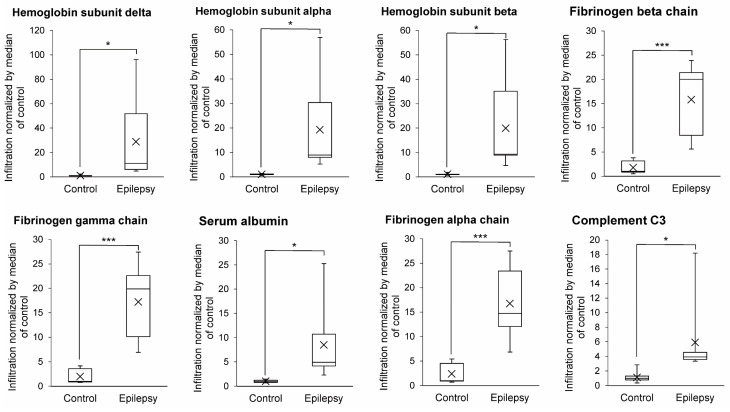
Infiltration of blood proteins in epileptic brain capillaries compared to controls. Brain capillaries were isolated from focal sites in six epilepsy patients and five normal brains; tryptic digests were produced and subjected to SWATH analysis. Blood proteins are defined as the molecules whose tissue specificity in the Uniprot database is classified as ”blood” or “plasma”. Among such molecules, the proteins showing the average ratio (Epilepsy/Control) > 5 and BH-adjusted *p* value < 0.05 are shown in this table. The band inside the box represents the median, and the bottom and top of the boxes indicate the first and third quartiles, respectively. The whiskers indicate the minimum and maximum values of the protein levels. X plots show the average in each group. The data are normalized by median of the infiltration in control. * BH-adjusted *p* value < 0.05 and *** BH-adjusted *p* value < 0.001, significantly different between epilepsy and control groups.

**Figure 5 pharmaceutics-13-02122-f005:**
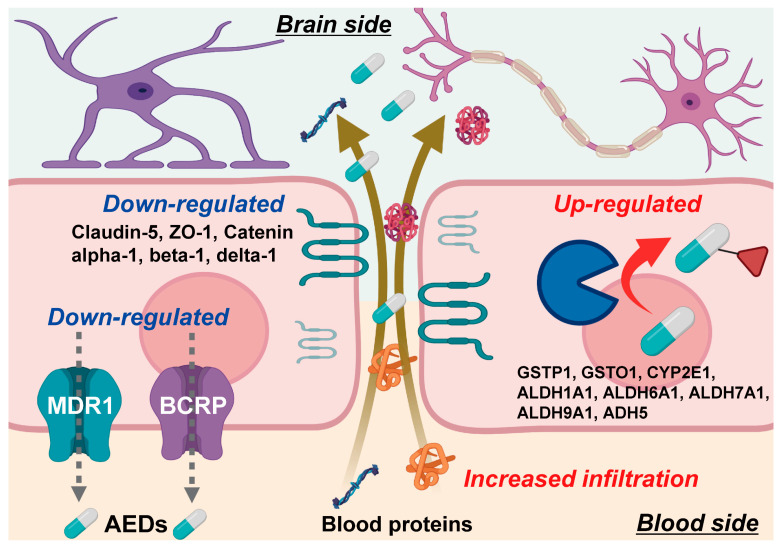
Summarized illustration of molecular mechanisms at the BBB in focal site of epileptic patient. These molecular mechanisms were suggested on the basis of the present SWATH results, but the functional demonstration will be needed in future.

**Table 1 pharmaceutics-13-02122-t001:** Information of brain tissues used for the present study.

Human Donor No.	Age	Gender	Pathological Diagnosis for Brain	AED Treatment	Response to AED Treatment	Brain Region Used
Patient 1	15 years	Male	Epilepsy	Clonazepam, Oxcarbazepine, Valproate	Refractory	Focal site
Patient 2	9 years	Male	Epilepsy	Levetiracetam, Lacosamide	Refractory	Focal site
Patient 3	17 years	Male	Epilepsy	Valproate, Lacosamide	Refractory	Focal site
Patient 4	19 years	Male	Epilepsy	Lacosamide, Topiramate	Refractory	Focal site
Patient 5	7 years	Female	Epilepsy	Oxcarbazepine, Phenobarbitol, Levetiracetam, Lacosamide, Clonazepam	Refractory	Focal site
Patient 6	18 years	Male	Epilepsy	No information	Refractory	Focal site
Control 1	16 years	Male	Normal	-	-	Cortex
Control 2	38 years	Male	Normal	-	-	Cortex
Control 3	26 years	Male	Normal	-	-	Cortex
Control 4-1	69 years	Male	Normal	-	-	Cortex
Control 4-2				-	-	White matter

All epilepsy patients had severe seizure disorder refractory to anti epileptic medications since all patients underwent surgery to try to cure the seizures. Brain capillaries were isolated from brain focal site in epilepsy patients or normal cortex/white matter in non-epileptic donors, and used for the SWATH analysis.

**Table 2 pharmaceutics-13-02122-t002:** Detected transporters and metabolizing enzymes which transport and metabolize anti-epileptic drugs.

Molecular Names	Anti-Epileptic Drugs Transported or Metabolized	References
**Transporters**		
MDR1 (efflux)	felbamate, phenobarbital, carbamazepine, lamotrigine, phenytoin, topiramate, levetiracetam oxcarbamazepine, acetazolamide, tiagabine	Nakanishi H et al., 2013 [7] Rizzi M et al., 2002 [27] Potschka H et al., 2002 [28] Clinckers R et al., 2005 [29] Crowe A et al., 2006 [30]
BCRP (efflux)	phenobarbital, clobazam, zonisamide, gabapentin, tiagabine, levetiracetam	Nakahashi H et al., 2013 [7]
MCT8 (efflux)	phenytoin	Jomura R et al., 2021 [25]
LAT1 (influx)	gabapentin, pregabalin	Dickens D et al., 2013 [31] Takahashi Y et al., 2018 [32]
**Metabolizing enzymes**		
Cytochrome P450 (CYP2E1)	ethosuximide, felbamate, phenobarbital, carbamazepine, valproate	Depondt C et al., 2006 [10] Bachmann K et al., 2003 [33]
Epoxide hydrolase (EPHX1)	carbamazepine, phenobarbital, phenytoin	Depondt C et al., 2006 [10]
Carbonyl reductase (CBR1,3)	oxcarbamazepine (activated by CBR)	Malatkova P et al., 2014 [34]
Aldehyde dehydrogenase (ALDH1A1, 2, 3A2, 6A1, 7A1, 9A1)	felbamate	Kapetanovic I et al., 2002 [35]
Alcohol dehydrogenase (ADH5)	felbamate	Di L et al., 2021 [36]
Glutathione S-transferase (GSTP1, O1, M3, K1, MGST2)	phenytoin, valproate, carbamazepine	Depondt C et al., 2006 [10] Tang W et al., 1996 [11] Yip V et al., 2017 [12]

The AED transporters and metabolizing enzymes detected by SWATH analysis are listed in this table. The substrates (AEDs) of the corresponding transporters and metabolizing enzymes are listed on the basis of literature search.

## Data Availability

Not applicable.

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
