# Peer review of "An Atlas of the Quantitative Protein Expression of Anti-Epileptic-Drug Transporters, Metabolizing Enzymes and Tight Junctions at the Blood–Brain Barrier in Epileptic Patients"

_pharmaceutics, 2021, doi:10.3390/pharmaceutics13122122_

Round 1

Reviewer 1 Report

Overall, the submitted manuscript has two aspects. One is the analytical for the quantitative determination of selected proteins that affect the permeability of antiepileptic drugs in the BBB. This part is a key part of the whole MS and is very, very beneficial. I really appreciate this part of the work! The second aspect is the clinical relevance of the studied proteins in relation to the response to treatment with a particular drug. Unfortunately, the authors did not manage this part well. Respectively, the interconnection of both of these parts is not managed here.

Major points:

  1. You try to draw pharmacokinetic and clinical conclusions from your purely analytical results without having any factual or experimental evidence. Either expand and supplement clinically relevant information or limit the content of the work and limit its interpretation to the analysis of the expression of the studied proteins and the measurement of BBB integrity with respect to blood protein permeability.
  2. The number of patients in the study is too small (6!) to make any generalizations.
  3. The characteristics of the patients enrolled in the study are too superficial: What AED were they treated? How long have they been treated? What was their response to treatment?
  4. Do the selected patients represent a homogeneous group with respect to the intended analysis? Can you demonstrate it?

  1. In your study, you focus on a wide range of genes, including ABC transporters. Why do you study only two of them, namely ABCB1 and ABCG2? Why not study others that can also interact with AEDs (members of ABCA subfamily, ABCB subfamily, and particularly ABCC subfamily?)

  1. I see the great benefit of this work in its analytical part, especially in the application of SWATH analysis. Unfortunately, the authors deal with this topic only marginally, both in the sections Introduction and Discussion. In contrast, they pay unnecessary attention to the clinical aspects of their results without having any experimental justification / results for them.

  1. You have studied the expression of proteins in the BBB that affect the effectiveness of the AED or may lead to resistance to the AED. However, your results showed the opposite: a reduced BBB function (reduced expression of ABCB1 and ABCG2 and a further reduction in its integrity). Can you address these results in broader context?

Minor points:

The description for Figure 4 is wrong! This is not “the expression” of blood proteins, but their penetration or infiltration…

Author Response

[Response to comment 1]

Thank you very much for your careful evaluation. As you mention, we are going to draw the pharmacokinetic and clinical conclusions from our analytical results without having any factual or experimental evidence. To tone down it, we added the following sentences in the conclusion section and Figure 5.

[Section 5]

… The present study has only revealed protein expression levels and it is not yet known whether these changes in expression levels affect the brain pharmacokinetics and pharmacological effects of antiepileptic drugs. Further experimental justification is required.

[Figure 5 legend]

Summarized illustration of molecular mechanisms at the BBB in focal site of epileptic patient. These molecular mechanisms were suggested on the basis of the present SWATH results, but the functional demonstration will be needed in future.

[Response to comment 2]

Thank you very much for your valuable comment. According to your comment, we added the discussion as follows;

[Section 4.]

A limitation of the present study is the small number of donors. Therefore, it cannot be said that all types of epilepsy patients are covered. In terms of age, patients with epilepsy tend to be younger than controls: Control 1 and patients with epilepsy 3, 4 and 6 overlap in age; for the molecules shown in Figures 1 to 4, there is no significant difference in expression between control 1 and older control donors (donors 2, 3 and 4) (Table S1). Therefore, the quantitative differences between epilepsy patients and controls shown in this study may be due to the pathogenesis of epilepsy. Only epileptic patient 5 was female, and the expression of GSTP1 was lower than in male epileptic patients (Table S1). GSTP1 is expressed in the liver at lower levels in females than in males [46]. Therefore, in BBB, the amount of GSTP1 may be lower in females. Further studies are needed to increase the number of samples and to equalize the age range and gender between epilepsy and control groups.

[Response to comment 3]

Thank you very much for your valuable comment. According to your comment, we added the clinical information about AED treatment in Table 1.

[Response to comment 4]

Because we cannot say that it is a homogenous group, we added the following sentences in discussion section.

[Section 4.]

A limitation of the present study is the small number of donors. Therefore, it cannot be said that all types of epilepsy patients are covered.

[Response to comment 5]

Thank you very much for your valuable comments. ABCA and B families (except for ABCB1) are basically not interacted with AEDs. ABCC families were not detected in the present SWATH study. It indicates that the protein expression levels of ABCC families are extremely small in the isolated human brain capillaries. Based on our protein expression level data, ABCB1 and ABCG2 are main transporters for AEDs at human BBB.

[Response to comment 6]

Thank you very much for your valuable comments. We added the following sentences.

[Section 4.]

… MDR1 at the BBB has been thought to be involved in drug resistance, as previous studies in rodents have shown that MDR1 expression limits the brain transfer of antiepileptic drugs. However, in humans, there is little data to suggest that MDR1 is involved in antiepileptic drug resistance. Immunostaining experiments have shown that the signal intensity of MDR1 is increased, but the specificity and quantitative performance of the antibody is limited. In this study, we applied the SWATH method, which has excellent specificity and quantitative accuracy, and for the first time we were able to accurately quantify the expression levels of MDR1 and BCRP. Therefore, unlike rodents, MDR1 and BCRP do not seem to have a significant effect on antiepileptic drug resistance in humans. Tight junctions were weakened in the focal area of human epilepsy, in agreement with reports in rodents [13,14]. Weakened tight junctions allow blood components to infiltrate the brain, and these have been reported to increase the frequency of epileptic seizures [13]. Thus, the brain transfer of antiepileptic drugs may be also increased, but the degree to which they induce convulsive seizures could be also increased, which may apparently indicate a weaker effect of antiepileptic drugs.

[Section 5]

… The present study has only revealed protein expression levels and it is not yet known whether these changes in expression levels affect the brain pharmacokinetics and pharmacological effects of antiepileptic drugs. Further experimental justification is required.

[Figure 5 legend]

Summarized illustration of molecular mechanisms at the BBB in focal site of epileptic patient. These molecular mechanisms were suggested on the basis of the present SWATH results, but the functional demonstration will be needed in future.

[Response to comment 7]

We added the following sentences and Figure 5 to address the results in broader context.

[Section 4.]

… MDR1 at the BBB has been thought to be involved in drug resistance, as previous studies in rodents have shown that MDR1 expression limits the brain transfer of antiepileptic drugs. However, in humans, there is little data to suggest that MDR1 is involved in antiepileptic drug resistance. Immunostaining experiments have shown that the signal intensity of MDR1 is increased, but the specificity and quantitative performance of the antibody is limited. In this study, we applied the SWATH method, which has excellent specificity and quantitative accuracy, and for the first time we were able to accurately quantify the expression levels of MDR1 and BCRP. Therefore, unlike rodents, MDR1 and BCRP do not seem to have a significant effect on antiepileptic drug resistance in humans. Tight junctions were weakened in the focal area of human epilepsy, in agreement with reports in rodents [13,14]. Weakened tight junctions allow blood components to infiltrate the brain, and these have been reported to increase the frequency of epileptic seizures [13]. Thus, the brain transfer of antiepileptic drugs may be also increased, but the degree to which they induce convulsive seizures could be also increased, which may apparently indicate a weaker effect of antiepileptic drugs.

[Response to minor comment]

The description of Figure 4 was corrected from “the expression” to “infiltration”.

Reviewer 2 Report

The authors assessed the quantitative Protein Expression of Anti-Epileptic-Drug Transporters, Metabolizing Enzymes and Tight Junctions at the Blood-Brain Barrier in Epileptic Patients, which is quite interesting and adding new information towards the clinical management of epilepsy. I have some concerns, which are given below:

  • In Table 1, the age of the control group is quite higher as compared to the epileptic group. This was not justified regarding its effect on the expression of proteins.
  • Does the difference in the brain region affect the quantity of expression of the proteins?
  • Why is control group number 4 divided into two subgroups?
  • As the author used the quantitative approach, do they perform, sensitivity analysis for the data analysis?
  • It would be helpful if the author adds the study limitation and its effect on the results.
  • In addition, it is advisable to add the practice implication of these findings.
  • Supplement the introduction section with more up to date information specially the role of polypharmacy in seizure management and  diseases that progress to full-blown epilepsy (PMID: 34768742; PMID: 33316769)

Author Response

[Response to comment 1]

Thank you very much for your valuable comment. According to your comment, we added the discussion as follows;

[Section 4.]

A limitation of the present study is the small number of donors. Therefore, it cannot be said that all types of epilepsy patients are covered. In terms of age, patients with epilepsy tend to be younger than controls: Control 1 and patients with epilepsy 3, 4 and 6 overlap in age; for the molecules shown in Figures 1 to 4, there is no significant difference in expression between control 1 and older control donors (donors 2, 3 and 4) (Table S1). Therefore, the quantitative differences between epilepsy patients and controls shown in this study may be due to the pathogenesis of epilepsy. Only epileptic patient 5 was female, and the expression of GSTP1 was lower than in male epileptic patients (Table S1). GSTP1 is expressed in the liver at lower levels in females than in males [46]. There-fore, in BBB, the amount of GSTP1 may be lower in females. Further studies are needed to increase the number of samples and to equalize the age range and gender between epilepsy and control groups.

[Response to comment 2 and 3]

Thank you very much for your valuable comment. According to your comment, we added the discussion as follows;

[Section 4.]

To investigate whether there are regional differences in the expression levels of the BBB, we isolated blood vessels from white matter and cortex for control 4 and performed SWATH analysis. The results showed that there was no significant difference in the expression of molecules in Figures 1 to 4 between white matter and cortical BBB (Table S1).

[Response to comment 4]

Sensitivity analysis for the data analysis is over-description in the present manuscript, and so we do not perform it.

[Response to comment 5]

Thank you very much for your valuable comment. According to your comment, we added the discussion as follows;

[Section 4.]

A limitation of the present study is the small number of donors. Therefore, it cannot be said that all types of epilepsy patients are covered. In terms of age, patients with epilepsy tend to be younger than controls: Control 1 and patients with epilepsy 3, 4 and 6 overlap in age; for the molecules shown in Figures 1 to 4, there is no significant difference in expression between control 1 and older control donors (donors 2, 3 and 4) (Table S1). Therefore, the quantitative differences between epilepsy patients and controls shown in this study may be due to the pathogenesis of epilepsy. Only epileptic patient 5 was female, and the expression of GSTP1 was lower than in male epileptic patients (Table S1). GSTP1 is expressed in the liver at lower levels in females than in males [46]. Therefore, in BBB, the amount of GSTP1 may be lower in females. Further studies are needed to increase the number of samples and to equalize the age range and gender between epilepsy and control groups.

We have previously reported that MDR1 protein expression at the BBB is increased in mouse models of epilepsy [41], but in contrast, it is decreased in patients with epilepsy (Figure 1). Table 1 lists the medications that epilepsy patients have been taking. These have not been reported to reduce MDR1 protein expression. Although the different age range of the epilepsy and control groups may have an effect, the expression of MDR1 was lower in the epilepsy group when compared between control 1 and epilepsy patients of the same age (donors 3, 4 and 6). In young adulthood, the expression of MDR1 in the BBB is highly variable in mice, but not in primates [47,48]. Although the detailed mechanisms are unknown, it may be likely that differences in the regulation of MDR1 expression between mice and primates result in the contrasting changes in expression levels at the BBB in mice and humans with epilepsy.

[Response to comment 6]

Thank you very much for your valuable comment. The practice implication of these findings is important, but the other reviewer pointed out that we pay unnecessary attention to the clinical aspects of the protein quantification data without having any experimental justification / results for them. Therefore, we do not add more practice implication of these findings.

[Response to comment 7]

Thank you very much for your valuable comment. We supplemented the introduction section with the two papers (PMID: 34768742; PMID: 33316769) you proposed, as follows;

[1. Introduction section]

… Post-traumatic epilepsy as a comorbidity of traumatic brain injuries is also becoming a universal challenge for brain health due to the increasing incidence of brain trauma [1]. ….., although a variety of combination pharmacotherapy of multiple drugs was investigated [3]. …

Reviewer 3 Report

The authors Sato et al., elucidate an important issue about epilepsy research. They find a novel and unbiased strategy to quantify the protein expression of AED-related proteins in cerebral vessels of epilepsy patients. By the SWATH proteomics method, they obtain a clear and accurate quantification of AED transporters, enzymes, and tight junction molecules at BBB. This report open a new window on potential increased infiltration of blood proteins by tight junctions at the BBB in the focal site of epilepsy. This is a well-written research paper. For the mentioned reasons, the manuscript needs a general vivid cartoon, which shows how the AED-related molecules are functionally related to each other and graphically depicts the BBB in the focal site of epilepsy.

The article may be accepted for publication after minor revision.

Author Response

[Response to comment 1]

Thank you very much for your valuable comment. According to your comment, we added the cartoon (Figure 5) to show how the AED-related molecules are functionally related to each other and graphically depict the BBB in the focal site of epilepsy.

Reviewer 4 Report

The work is of high quality and the manuscript well documented. A nice demonstration of the interest of SWATH analysis for proteomics. All the conclusions are supported by carefully validated results. The authors are aware that in vivo analyses will now be needed to validate the involvement of the "up-regulated" enzymes in drug resistance.

Author Response

[Response to comment 1]

Thank you very much for your valuable comment. We added the following sentences in the conclusion section and Figure 5.

[Section 5]

… The present study has only revealed protein expression levels and it is not yet known whether these changes in expression levels affect the brain pharmacokinetics and pharmacological effects of antiepileptic drugs. Further experimental justification is required.

[Figure 5 legend]

Summarized illustration of molecular mechanisms at the BBB in focal site of epileptic patient. These molecular mechanisms were suggested on the basis of the present SWATH results, but the functional demonstration will be needed in future.

Round 2

Reviewer 1 Report

Satisfied with the adjustment you did!